# Yearly Incidence of Stroke and Bleeding in Atrial Fibrillation with Concomitant Hyperthyroidism: A National Discharge Database Study

**DOI:** 10.3390/jcm11051342

**Published:** 2022-02-28

**Authors:** Juqian Zhang, Arnaud Bisson, Grégoire Fauchier, Alexandre Bodin, Julien Herbert, Pierre Henri Ducluzeau, Gregory Y. H. Lip, Laurent Fauchier

**Affiliations:** 1Liverpool Centre for Cardiovascular Science, Liverpool Heart & Chest Hospital, University of Liverpool, Liverpool L7 8TX, UK; juqian.zhang@liverpool.ac.uk; 2Service de Cardiologie, Centre Hospitalier Universitaire et Faculté de Médecine, Université de Tours, 37044 Tours, France; arnaud.bisson37@gmail.com (A.B.); alexandrebodin.mail@gmail.com (A.B.); j.herbert@chu-tours.fr (J.H.); laurent.fauchier@univ-tours.fr (L.F.); 3Service de Médecine Interne, Unité d’Endocrinologie Diabétologie et Nutrition, Centre Hospitalier Universitaire et Faculté de Médecine, Université de Tours, 37044 Tours, France; gfauchier@gmail.com (G.F.); ph.ducluzeau@chu-tours.fr (P.H.D.); 4Institut National de Recherche pour l’Agriculture, l’Alimentation et l’Environnement, Unité Mixte de Recherche Physiologie de la Reproduction et des Comportements, 37380 Nouzilly, France

**Keywords:** atrial fibrillation, hyperthyroidism, ischaemic stroke, bleeding risk

## Abstract

Background: Hyperthyroidism is associated with atrial fibrillation (AF), and the latter is a major risk factor for stroke. Aim: We aimed to investigate the yearly incidence of stroke and bleeding in AF patients with and without concomitant hyperthyroidism from the French National Hospital Discharge Database. Methods: Admissions with AF between January 2010 and December 2019 were retrospectively identified and retrieved from the French national database. Incidence rates of ischaemic stroke and bleeding were compared in AF patients with and without concomitant hyperthyroidism. The associations of risk factors with ischaemic stroke were assessed by Cox regression. Results: Overall 2,421,087 AF patients, among whom 32,400 had concomitant hyperthyroidism were included in the study. During the follow-up (mean: 2.0 years, standard deviation SD: 2.2 years), the yearly incidence of ischaemic stroke was noted to be 2.6 (95% confidence interval CI: 2.5–2.8) in AF patients with concomitant hyperthyroidism, and 2.3 (95%CI: 2.3–2.4) in non-thyroid AF patients. Hyperthyroidism was noted as an independent risk factor for ischaemic stroke (adjusted hazard ratio aHR: 1.133, 95%CI: 1.080–1.189) overall, particularly within the first year of hyperthyroidism diagnosis (aHR 1.203, 95%CI 1.120–1.291), however, the association became non-significant in subsequent years (aHR 1.047, 95%CI 0.980–1.118). Major bleeding incidence was lower in the hyperthyroid AF group in comparison to the non-thyroid AF group (incidence ratio: 5.1 vs. 5.4%/year, *p* < 0.001). The predictive value of CHA_2_DS_2_VASc and HAS-BLED scores for ischaemic stroke and bleeding events, respectively, did not differ significantly between AF patients with or without concomitant hyperthyroidism. Conclusions: Hyperthyroidism seems to be an independent risk factor of ischaemic stroke in AF patients, particularly within the first year of hyperthyroidism diagnosis.

## 1. Introduction

Hyperthyroidism, a common endocrine disorder, affects 0.5% to 2% of the general population [1]. Atrial fibrillation (AF) is the most common cardiac condition associated with hyperthyroidism; its prevalence was estimated to be 10% to 25% in overtly hyperthyroid patients as compared to 1.5–2% in the general population [2]. Hyperthyroidism is commonly associated with AF [3]. New-onset AF has also been associated with more than double the risk of hyperthyroidism, especially among middle-aged men [4]. In a meta-analysis of eleven cohort studies, hyperthyroidism was also associated with a pooled hazard ratio (HR) of 1.35 (95% confidence interval CI: 1.03–1.75) for stroke and 1.20 (95%CI: 1.00–1.46) for cardiovascular mortality [5]. In its subgroup analyses, however, the association was insignificant in those ≥65 years of age or those with high cardiovascular risk (including AF) (5). These findings suggest a diminishing association between hyperthyroidism and stroke with increasing overall stroke risk factors in the AF population.

### Aims

We aimed to retrospectively assess the yearly incidence of ischaemic stroke, major bleeding events, all-cause mortality and cardiovascular mortality in the AF patients with and without concomitant hyperthyroidism from the French National Hospital Discharge Database. We also aim to identify the independent risk factors for ischaemic stroke in AF patients who were hospitalised in France, as well as to evaluate the predictive accuracy of CHA_2_DS_2_VASc and HAS-BLED scores in AF patients with and without concomitant hyperthyroidism.

## 2. Methods

This longitudinal national study was performed on the National Hospitalization Discharge Database (PMSI, abbreviation for the Programme de Medicalisation des Systemes d’Information in French). Discharge diagnoses nationwide have been coded using the 10th revision of the International classification of diseases (ICD-10) in PMSI since 1996. Data for all patients admitted with AF in France from January 2010 to December 2019 were identified and retrieved from the PMSI following the updated version of the ICD10 for the years 2010–2019.

The medical information available from the database has been anonymised and coded, therefore, patient consent was not required. The study was approved by the institutional review board of the Pole Coeur Thorax Vaisseaux from the Trousseau University Hospital (Tours, France) on 1 December 2015. The data collection and data management of the study were approved by the Conseil National de l’Informatique et des Libertés which safeguards the confidentiality and anonymity of the process (authorization no. 1749007).

We evaluated the CHA_2_DS_2_VASc score and a proxy of the HAS-BLED score (including all the risk factors except labile INR which was inaccessible from the database). Hyperthyroidism was identified with its specific ICD-10 code (E05) with the date of its first diagnosis in the database. Concomitant AF and hyperthyroidism, defined with an interval less than 45 days between first diagnoses of hyperthyroidism and AF, were compared with patients with AF and non-thyroid. Patients with previous hyperthyroidism more than 45 days before diagnosis of AF were excluded (Figure 1).

The endpoints of ischaemic stroke (ICD code I63), major bleeding and intracranial haemorrhage (ICH) were investigated from January 2010 in the study, and the individual follow-up for each candidate started at the time of first documented AF diagnosis until each specific outcome or the last recorded follow-up in the absence of an outcome. Information on outcomes during follow-up was obtained by analysing the PMSI codes for each patient. Major bleeding was defined as bleeding with new anaemia, or with transfusion of at least 1 unit of blood, or symptomatic bleeding in a critical organ or body area (e.g., intracranial, retroperitoneal, intra-articular, or pericardial) or bleeding that causes death [6].

### Statistical Analysis

Qualitative variables were described using counts and percentages, while continuous quantitative variables were described as mean and standard deviation or median and interquartile range. Comparisons were made using parametric or nonparametric tests when appropriate: The Wilcoxon signed rank and Kruskal–Wallis tests were used for comparing values between 2 independent groups, and the χ^2^ test was used to compare categorical data. Incidence rates (IR) with 95% Confidence Interval (95%CI) were calculated for ischaemic stroke and bleeding events according to the presence of concomitant hyperthyroidism and by multivariable Cox regression models to calculate the relative hazard ratio (HR) and 95%CI for each clinical variable. A proportional hazards model was used to identify independent risk factors associated with the occurrence of each clinical outcome.

Receiver operating characteristic (ROC) curves were constructed, and Harrell C indexes (i.e., area under the curve) were calculated to investigate the predictive value of CHA_2_DS_2_VASc and HAS-BLED scores in patients with and without hyperthyroidism; the two curves were compared using the DeLong test.

In all analyses, *p* < 0.05 was considered statistically significant. Analyses were performed using Enterprise Guide 7.1, (SAS Institute Inc., SAS Campus Drive, Cary, NA, USA), USA and STATA version 16.0 (Stata Corp, College Station, TX, USA).

## 3. Results

From January 2010 to December 2019, 2,435,541 adults (age ≥ 18 years) were hospitalised with a diagnosis of AF (I48) as the principal diagnosis (i.e., the condition justifying hospital admission), a related diagnosis (i.e., potential chronic disease or health state during the hospital stay), or a significantly associated diagnosis (i.e., comorbidity or associated complication) (Figure 1). This analysis included 2,421,087 AF patients (mean age 77.2 ± 12.1 years; 52.9% women) of which 32,400 (1.3%) had concomitant hyperthyroidism. The interval between the first diagnoses of AF and hyperthyroidism was 1.4 ± 9.5 days (median 0, IQR 00).

Patients with hyperthyroidism had some different characteristics compared to those without, including a higher prevalence of female sex, heart failure and dilated cardiomyopathy (DCM), and lower prevalence of comorbidities, such as coronary artery disease, previous myocardial infarction, vascular disease and cancer (Table 1 and Appendix A).

### 3.1. Hyperthyroidism and Ischaemic Stroke

During the follow-up (mean 2.0 ± 2.2 years, median 1.0 IQR 0.1–3.3 years), the accumulative new-onset ischaemic stroke cases were 1669 (Incidence Rate IR: 2.6%/year) in patients with and 30,731 (IR: 2.3%/year) in patients without hyperthyroidism. Figure 2 shows the incidences during the first year of follow-up, and Figure 3 shows the incidences beyond the first year of follow-up. On multivariable Cox regression analysis, the presence of hyperthyroidism (HR 1.133, 95%CI 1.080–1.189, *p* < 0.0001) was independently associated with ischaemic stroke in the overall cohort (Table 2) This risk was particularly evident within the first year of hyperthyroidism diagnosis (HR 1.203, 95%CI 1.120–1.291), with a nonsignificant association beyond 1 year (HR 1.047, 95%CI 0.980–1.118).

The yearly incidences of stroke in AF patients stratified by sex, the presence or absence of concomitant hyperthyroidism, and the CHA_2_DS_2_VASc score are presented in Appendix A for the whole follow-up analysis, as well as in Figure 2 for yearly incidences in the first year of follow-up and Figure 3 for yearly incidences after the first year of follow-up, respectively. The incidence of ischaemic stroke was higher in the first year after AF diagnosis than in the subsequent follow-up: yearly incidence 3.24% (95% CI 3.21–3.26) in the first year following the diagnosis of AF vs. 1.95% (95% CI 1.93–1.96) beyond. The overall incidence of ischaemic stroke was higher in patients with hyperthyroidism at baseline and the difference was more marked in women than in men (Appendix A). The overall risk of ischaemic stroke was 1.20 in HR (95%CI 1.12–1.29) in hyperthyroidism as compared to non-hyperthyroidism in the first year and was significant in both men and women (Appendix A). Beyond the first year, the risk of ischaemic stroke was slightly lower in male AF patients with hyperthyroidism compared to those without (overall HR 0.86, 95%CI 0.76–0.97), and there was no significant difference in the risk of ischaemic stroke between female patients with and without concomitant hyperthyroidism (overall HR 1.06, 95%CI 0.98–1.15) (Figure 4 and Appendix A). An increasing CHA_2_DS_2_VASc score was consistently associated with a higher incidence of ischaemic stroke during follow-up in both groups of patients with and without hyperthyroidism.

Analysis of ROC curves showed no significant difference in the CHA_2_DS_2_-VASc score for the prediction of ischaemic stroke in patients with and without hyperthyroidism: 0.600 (95%CI 0.586–0.613) vs. AUC 0.600 (95%CI 0.597–0.601), respectively (*p* = 0.94 for DeLong test) (Figure 5).

### 3.2. Hyperthyroidism and Mortality

Compared with ischaemic stroke, no clear differences were observed in the yearly incidences of all-cause mortality and cardiovascular mortality in AF patients with or without concomitant hyperthyroidism, respectively, although a similar trend of significantly higher mortality rates was noted within the first year of hyperthyroidism diagnosis as compared to further beyond (Table 3). Hyperthyroidism was associated with slightly lower all-cause and cardiovascular mortality within the first year in follow-up, and slightly lower risk for all-cause death was noted in AF with concomitant hyperthyroidism over the entire follow-up period; however, no significant association was observed for cardiovascular death during the whole follow-up period (Table 4).

### 3.3. Bleeding Events and HAS-BLED Score

AF patients with concomitant hyperthyroidism had a lower incidence of major bleeding (*n* = 3085, IR 5.1%/year) compared to those without (*n* = 238,565, IR 5.4%/year, *p* < 0.0001). Analysis of ROC curves for HAS-BLED score predicting major bleeding in patients with and without hyperthyroidism showed that the predictive performance was good with an AUC 0.785 (95%CI 0.784–0.786) in patients without hyperthyroidism, and AUC 0.785 (95%CI 0.777–0.792) in patients with a history of hyperthyroidism (DeLong test, *p* = 0.99) (Figure 5).

Regarding ICH, patients with hyperthyroidism had a similar incidence of ICH (*n* = 777, IR 1.2%/year) compared to those without (*n* = 56,305, IR 1.2%/year, *p* = 0.80). Analysis of ROC curves for HAS-BLED score predicting ICH in patients with and without hyperthyroidism also showed that the predictive performance was good with an AUC 0.735 (95%CI 0.733–0.737) in patients without hyperthyroidism, and AUC 0.738 (95%CI 0.724–0.752) in patients with a history of hyperthyroidism (DeLong test, *p* = 0.70) (Figure 5).

## 4. Discussion

This is the largest published analysis using a national discharge database on the yearly incidence of stroke and bleeding in hospitalised AF patients with and without concomitant hyperthyroidism. It noted an increased incidence of ischaemic stroke in hospitalised AF patients with hyperthyroidism compared to those without, and the highest incidence was noted in the first year following the diagnosis of hyperthyroidism. Beyond the first year in follow-up, the risk of ischaemic stroke among all the AF patients with hyperthyroidism was not statistically different than in those without. The CHA_2_DS_2_-VASc score and HAS-BLED scores performed similarly well in patients with AF with and without hyperthyroidism for predicting ischaemic stroke and major bleeding. Thus, the HAS-BLED score is the only bleeding risk score tested in patients with AF and hyperthyroidism.

Compared with other hospital-based cohort studies [7,8], the current study is based on a “real-world” discharge dataset at a much larger scale. There were some differences in the demographics and cardiovascular risk profiles in the hyperthyroid AF patients as compared to non-thyroid AF patients in the current study. The baseline characteristics in the current study resemble that of Bruere et al. [8], although a higher CHA_2_DS_2_-VASc score (mean: 3.6 vs. 3.1), and lower previous ischaemic stroke rate (6% vs. 9%) were noted in the current cohort compared with theirs. A lower yearly incidence of ischaemic stroke was noted in the current study in comparison with the cohort of Chen et al. (2.6% vs. 7.6% in hyperthyroid AF, 2.3% vs. 3.6% in non-thyroid AF) [7]. The difference in stroke incidence could be due to younger subjects in whom hyperthyroidism might be a more prominent risk factor in an otherwise lower risk population for ischaemic stroke [9], as well as the possible difference in the prescription rate of oral anticoagulation between the studies. A Mendelian randomization study using genome-based data reported a causal association between TSH levels and free thyroxine and stroke, mainly mediated via AF [10]. The higher risk of stroke associated with hyperthyroidism was mainly seen in patients with intermediate risk (e.g., CHA2DS2-VASc score 4 group). There is no clear explanation for this finding, but one might suggest that the higher risk of stroke with hyperthyroidism may not be clinically significant when the intrinsic risk of stroke is low, whereas this effect is mitigated by other risk factors when the risk of stroke is high.

The finding of highest ischaemic stroke incidence within the first year of diagnosis of hyperthyroidism in the AF population was similar to a Danish population-based cohort where the highest risk for ischaemic stroke (adjusted HR: 2.70, 95%CI: 2.31–3.14) was in the first 3 months following the diagnosis of hyperthyroidism in a general population for whom the prevalence of AF at baseline was not reported [11]. Similarly, a recent Korean national database-derived study also identified an increased risk of thromboembolism within the first year of a hyperthyroidism-related AF diagnosis [12]. In another cohort which included 160 patients with hyperthyroidism-induced AF, the majority of ischaemic strokes occurred within the first 30 days of initial presentation of AF (73% vs. 80% in hyperthyroid AF vs. non-thyroid AF) [13]. Hyperthyroidism causes a proinflammatory, procoagulant state in acute phases [14], arterial remodelling, stiffness and atherosclerosis over longer terms [15]. The acute prothrombotic phase immediately following the diagnosis of hyperthyroidism might be a treatment window to prevent thromboembolism and other cardiovascular events [16,17]. The finding of the current study suggests the importance of prompt risk stratification for stroke and initiation of oral anticoagulation following the diagnosis of hyperthyroidism [18], as well as timely treatment of hyperthyroidism to prevent stroke [19]. Beyond the higher risk of ischaemic stroke in the first months, current results also contradict the idea that the risk of ischaemic stroke would be markedly lower (and OAC not needed) in patients with transient AF once hyperthyroidism has been appropriately managed, confirming results from smaller reports about AF and a presumed “temporary cause” [20].

Two recent retrospective nationwide database-derived studies using propensity score-matched analyses in the developed region in Asia had contradictory findings in the risk of thromboembolism associated with hyperthyroid AF [12,21]. Cautious interpretation of these findings, while acknowledging the inherent limitations and bias of retrospective analysis of coded information is advised. Further large-scale prospective matched-cohort studies which investigate and characterize both the short-term and the long-term stroke events of new-onset hyperthyroidism in AF patients would be desirable. Prospective trials investigating the effectiveness and feasibility of a clinical algorithm that promotes timely and accurate stroke risk assessment, initiation of OAC, and hyperthyroidism treatment for concomitant hyperthyroidism in AF patients with varying thromboembolic risk profiles could provide further guidance to clinical practice.

### Limitations

Due to its observational design using a discharge database with ICD coding for diagnosis, there was a lack of ascertainment of individual cases. There is a lack of information over the coherence of the hyperthyroidism diagnosis, as well as the clinical course of the condition. Abnormal thyroid function tests were not uncommon among patients with recent onset of AF; however, overt hyperthyroidism is rather uncommon over medium terms follow-up [22], thereby, subclinical or undiagnosed hyperthyroidism could have contributed to AF unnoticeably in a proportion of the cohort. The study adopted the cut-off interval as within 45 days between the first diagnosis of hyperthyroidism and AF for inclusion, however, some patients included in the cohort could have been affected by subclinical or undiagnosed hyperthyroidism for a more extended period before the inclusion. This limitation could potentially affect the interpretation of the relationship between the hyperthyroidism diagnosis and the observed clinical outcomes. Additionally, given the information on the initiation and efficacy of antithyroid treatment were also lacking, it is uncertain whether hyperthyroidism treatment could have reduced the difference in stroke incidence between the two groups beyond the first year of diagnosis. Given that the current study only included patients hospitalised with AF, patients with presumably milder symptoms or asymptomatic with AF could be missed from the study, possible selection bias exists and needs to be taken into consideration during the interpretation and extrapolation of the findings.

Another limitation of the study is its lack of information on the prescription, particularly concerning the initiation, duration, and effectiveness of the oral anticoagulation in the national database investigated. The delayed prescription and suboptimal dosing and adherence to anticoagulation could be a potentially important confounding factor for the increased stroke events observed within the first year of hyperthyroidism diagnosis. Furthermore, results may be different with direct oral anticoagulants that are associated with improved survival compared to vitamin K antagonists in patients with AF, irrespective of concomitant hyperthyroidism [18,23].

There is a major limitation in the evaluation process for the independent risk factors of ischaemic stroke due to its design using two unmatched cohorts from an ICD-coded database. Despite adopting multivariate Cox regression which incorporated all the clinical risk factors identified and accessible in the database in the risk factor analysis, the lack of direct information on the prescription and treatment efficacy of oral anticoagulation, the potential underdiagnosis and undertreatment of subclinical thyroid disorder, etc., could potentially introduce confounding bias and compromise the association between hyperthyroidism and ischaemic stroke.

The generalization of our findings to other populations and regions might also be limited given different ethnic compositions, level of healthcare service and prevalence of the relevant diseases and risk factors.

## 5. Conclusions

In this nationwide atrial fibrillation study, the highest incidence of ischaemic stroke was noted in hospitalised AF patients within the first year of hyperthyroidism diagnosis, during which period hyperthyroidism seemed to be an independent risk factor of ischaemic stroke. Beyond the timeframe, however, there was no independent contribution of hyperthyroidism to ischaemic stroke in hospitalised AF patients. No significant difference was noted between hospitalised AF patients with or without concomitant hyperthyroidism in the yearly incidence of all-cause mortality and cardiovascular mortality, respectively. Hospitalised AF patients with concomitant hyperthyroidism were noted with a lower incidence of major bleeding and similar incidence of ICH as compared to those without.

## Figures and Tables

**Figure 1 jcm-11-01342-f001:**
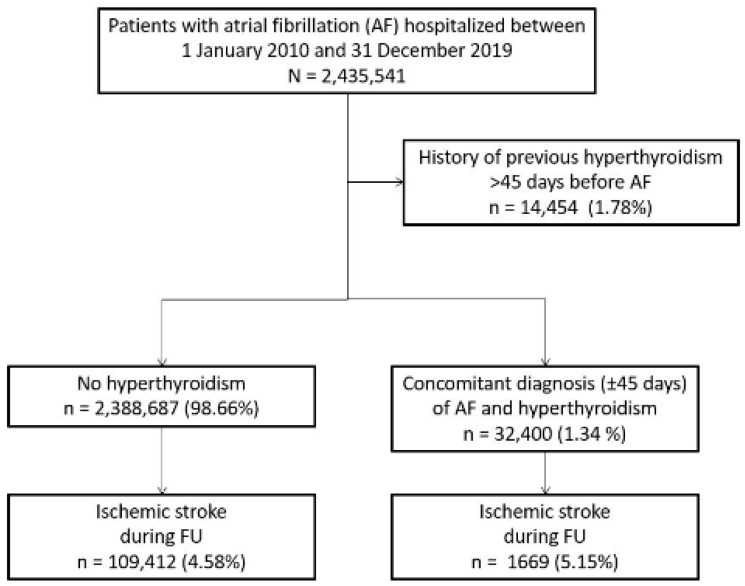
Flow chart of patients included in the study. FU: follow-up.

**Figure 2 jcm-11-01342-f002:**
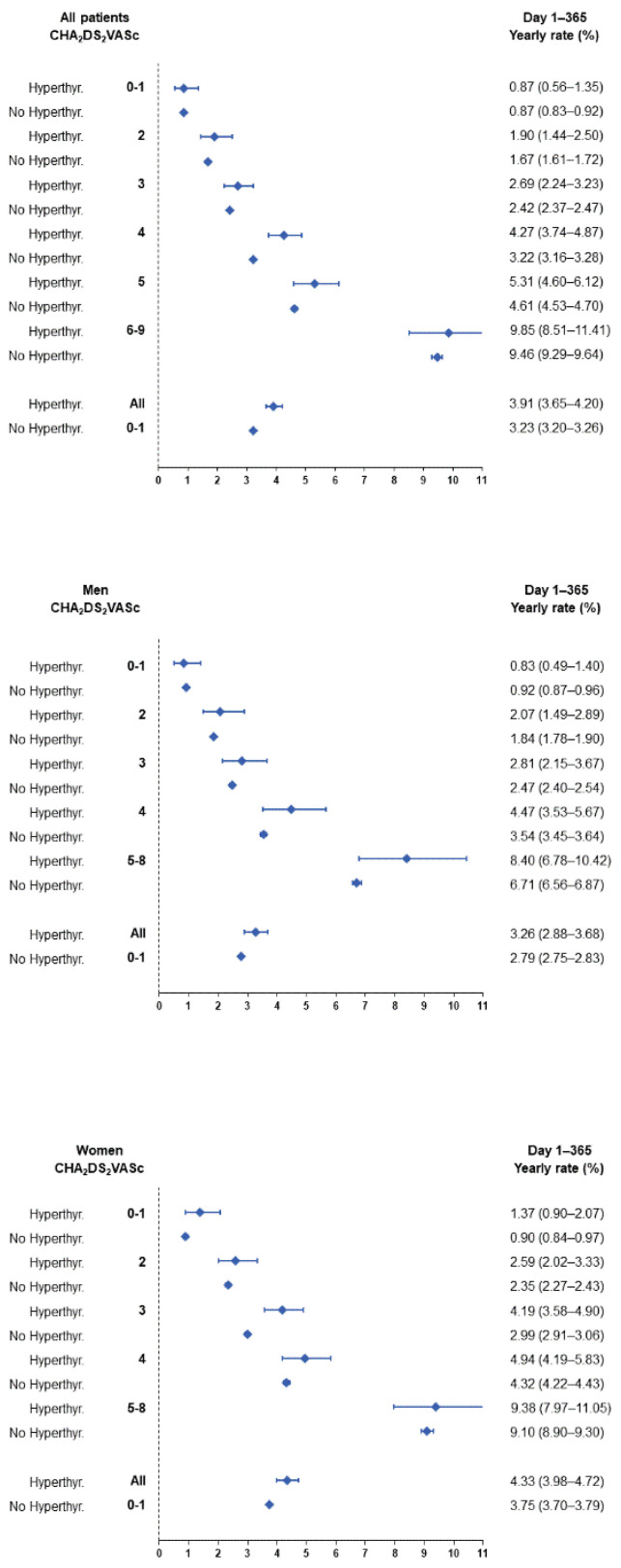
The yearly incidence of stroke in AF patients stratified by sex, CHA2DS2-VASc score and presence or absence of concomitant hyperthyroidism during the first year of follow-up.

**Figure 3 jcm-11-01342-f003:**
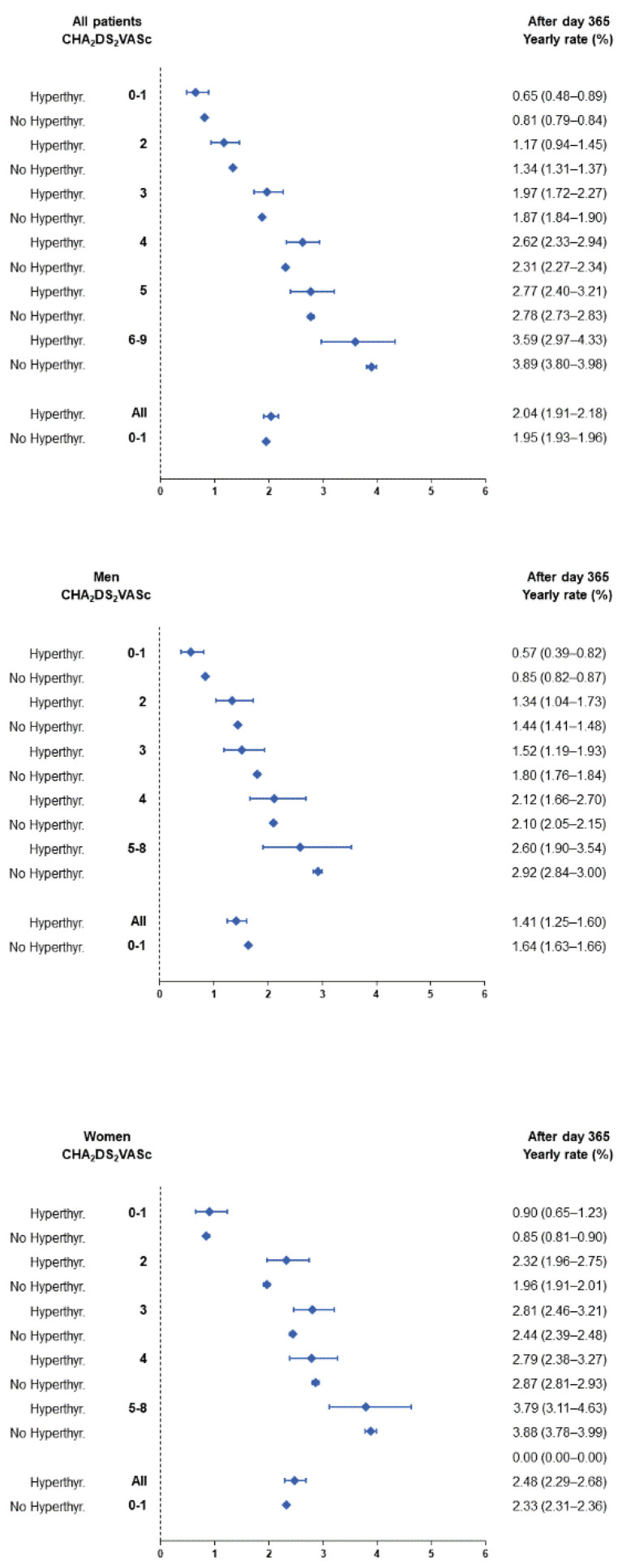
The yearly incidence of stroke in AF patients stratified by sex, CHA2DS2-VASc score and presence or absence of concomitant hyperthyroidism beyond the first year of follow-up.

**Figure 4 jcm-11-01342-f004:**
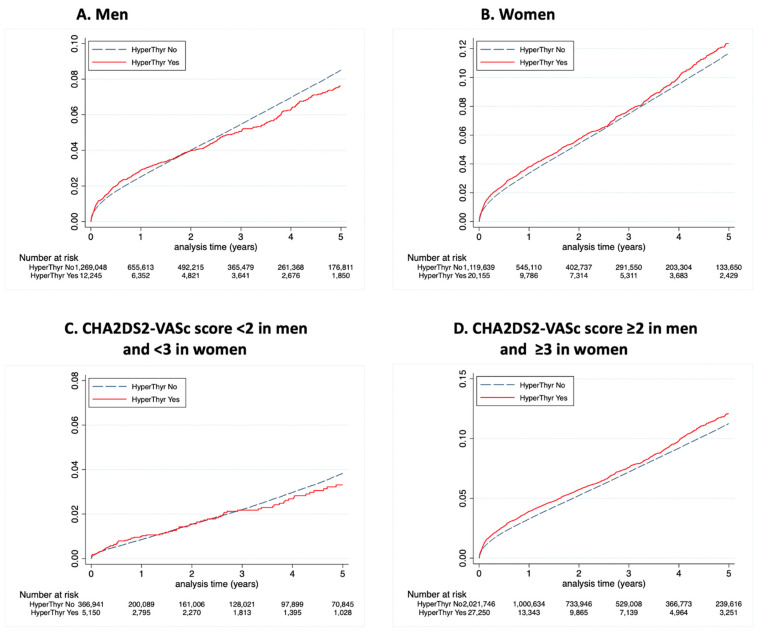
The cumulative incidence for ischaemic stroke in different sex groups and CHA2DS-VASc score. Cumulative incidences for ischaemic stroke in men (**A**), women (**B**), CHA2DS2-VASc score <2 in men and <3 in women (**C**), and CHA2DS2-VASc score ≥2 in men and ≥3 in women (**D**) with and without hyperthyroidism concomitant to AF diagnosis (interval <45 days between first diagnoses of AF and hyperthyroidism).

**Figure 5 jcm-11-01342-f005:**
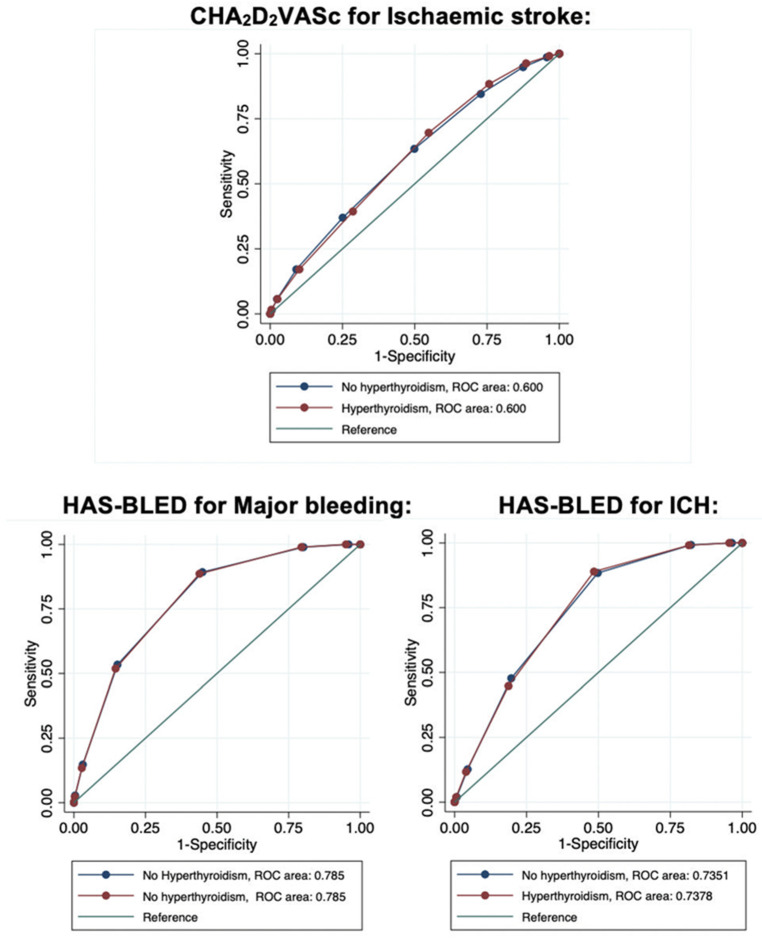
Receiver Operative Characteristic (ROC) curves in AF patients with and without hyperthyroidism for CHA2DS2-VASc score predicting ischaemic stroke (top panel, *p* for DeLong test = 0.94) and HAS-BLED score predicting major bleeding (lower left panel, *p* for DeLong test = 0.99) and intracranial haemorrhage ICH (lower right panel, *p* for DeLong test = 0.70).

**Table 1 jcm-11-01342-t001:** Baseline characteristics of patients with AF seen in French hospitals (2010–2019) according to the presence of concomitant hyperthyroidism or not.

	No Hyperthyroidism	Hyperthyroidism	*p*	Total
	(*n* = 2,388,687)	(*n* = 32,400)		(*n* = 2,421,087)
Age, years	77.2 ± 12.1	77.0 ± 12.5	0.01	77.2 ± 12.1
Sex (male)	1,269,048 (53.1)	12,245 (37.8)	<0.0001	1,281,293 (52.9)
CHA_2_DS_2_VASc score	3.5 ± 1.6	3.6 ± 1.6	<0.0001	3.5 ± 1.6
HAS-BLED score	2.4 ± 1.2	2.3 ± 1.2	<0.0001	2.4 ± 1.2
Charlson comorbidity index	3.5 ± 2.8	3.4 ± 2.7	0.34	3.5 ± 2.8
Frailty index	9.2 ± 9.1	10.5 ± 9.4	<0.0001	9.3 ± 9.1
Hypertension	1,428,261 (59.8)	19,496 (60.2)	0.17	1,447,757 (59.8)
Diabetes mellitus	498,572 (20.9)	6706 (20.7)	0.44	505,278 (20.9)
Heart failure	861,573 (36.1)	13,234 (40.8)	<0.0001	874,807 (36.1)
History of pulmonary oedema	59,076 (2.5)	725 (2.2)	0.01	59,801 (2.5)
Mitral regurgitation	139,496 (5.8)	2180 (6.7)	<0.0001	141,676 (5.9)
Aortic regurgitation	58,330 (2.4)	807 (2.5)	0.57	59,137 (2.4)
Aortic stenosis	146,768 (6.1)	1742 (5.4)	<0.0001	148,510 (6.1)
Previous endocarditis	10,537 (0.4)	110 (0.3)	0.01	10,647 (0.4)
Dilated cardiomyopathy	164,626 (6.9)	2720 (8.4)	<0.0001	167,346 (6.9)
Coronary artery disease	584,941 (24.5)	6798 (21.0)	<0.0001	591,739 (24.4)
Previous myocardial infarction	116,836 (4.9)	1354 (4.2)	<0.0001	118,190 (4.9)
Previous PCI	95,351 (4.0)	978 (3.0)	<0.0001	96,329 (4.0)
Previous CABG	78,453 (3.3)	713 (2.2)	<0.0001	79,166 (3.3)
Vascular disease	419,415 (17.6)	5003 (15.4)	<0.0001	424,418 (17.5)
Previous pacemaker or ICD	92,335 (3.9)	998 (3.1)	<0.0001	93,333 (3.9)
Previous ischaemic stroke	170,283 (7.1)	1953 (6.0)	<0.0001	172,236 (7.1)
Previous intracranial bleeding	51,496 (2.2)	582 (1.8)	<0.0001	52,078 (2.2)
Smoker	159,153 (6.7)	2136 (6.6)	0.61	161,289 (6.7)
Dyslipidaemia	513,732 (21.5)	6107 (18.8)	<0.0001	519,839 (21.5)
Obesity	339,579 (14.2)	4174 (12.9)	<0.0001	343,753 (14.2)
Alcohol-related diagnoses	120,241 (5.0)	1208 (3.7)	<0.0001	121,449 (5.0)
Abnormal renal function	162,550 (6.8)	2265 (7.0)	0.19	164,815 (6.8)
Lung disease	393,587 (16.5)	5458 (16.8)	0.08	399,045 (16.5)
Sleep apnoea syndrome	124,669 (5.2)	1323 (4.1)	<0.0001	125,992 (5.2)
COPD	223,243 (9.3)	3063 (9.5)	0.51	226,306 (9.3)
Liver disease	86,786 (3.6)	1105 (3.4)	0.03	87,891 (3.6)
History of hypothyroidism	168,451 (7.1)	2550 (7.9)	<0.0001	171,001 (7.1)
Inflammatory disease	141,804 (5.9)	1954 (6.0)	0.48	143,758 (5.9)
Anaemia	404,101 (16.9)	5647 (17.4)	0.01	409,748 (16.9)
Previous cancer	420,077 (17.6)	4843 (14.9)	<0.0001	424,920 (17.6)

Values are *n* (%) or mean ± SD. CABG = coronary artery bypass graft; COPD = chronic obstructive pulmonary disease; MI = myocardial infarction; PCI = percutaneous coronary intervention; SD = standard deviation.

**Table 2 jcm-11-01342-t002:** Predictors of ischaemic stroke during follow-up in AF patients.

	Univariate Analysis HR, 95%CI	*p*	Multivariable Analysis HR, 95%CI	*p*
Age, years	1.038 (1.038–1.039)	<0.0001	1.034 (1.033–1.035)	<0.0001
Sex (male)	0.721 (0.712–0.729)	<0.0001	0.863 (0.853–0.875)	<0.0001
Hypertension	1.279 (1.263–1.295)	<0.0001	1.052 (1.038–1.066)	<0.0001
Diabetes mellitus	1.148 (1.132–1.164)	<0.0001	1.185 (1.168–1.202)	<0.0001
Heart failure with congestion	1.095 (1.081–1.110)	<0.0001	1.053 (1.037–1.068)	<0.0001
History of pulmonary oedema	0.987 (0.944–1.033)	0.57	1.002 (0.957–1.050)	0.92
Mitral regurgitation	0.984 (0.959–1.010)	0.22	0.973 (0.947–0.999)	0.05
Aortic regurgitation	0.999 (0.961–1.038)	0.97	0.975 (0.937–1.015)	0.22
Aortic stenosis	1.060 (1.034–1.087)	<0.0001	1.009 (0.983–1.035)	0.50
Previous endocarditis	1.388 (1.279–1.506)	<0.0001	1.402 (1.290–1.522)	<0.0001
Dilated cardiomyopathy	0.940 (0.918–0.963)	<0.0001	1.053 (1.027–1.079)	<0.0001
Coronary artery disease	1.013 (0.999–1.027)	0.07	0.998 (0.981–1.015)	0.78
Previous myocardial infarction	1.122 (1.091–1.155)	<0.0001	1.029 (0.994–1.064)	0.11
Previous PCI	0.949 (0.918–0.980)	0.002	0.969 (0.934–1.005)	0.09
Previous CABG	0.914 (0.884–0.946)	<0.0001	0.956 (0.923–0.992)	0.02
Vascular disease	1.249 (1.230–1.267)	<0.0001	1.167 (1.146–1.188)	<0.0001
Previous pacemaker or ICD	1.012 (0.980–1.045)	0.47	0.915 (0.886–0.945)	<0.0001
Previous ischaemic stroke	5.207 (5.133–5.282)	<0.0001	4.681 (4.612–4.750)	<0.0001
Previous intracranial bleeding	2.588 (2.507–2.672)	<0.0001	1.719 (1.664–1.775)	<0.0001
Smoker	0.839 (0.817–0.861)	<0.0001	1.137 (1.105–1.170)	<0.0001
Dyslipidaemia	1.022 (1.008–1.037)	0.002	0.928 (0.914–0.942)	<0.0001
Obesity	0.765 (0.751–0.780)	<0.0001	0.875 (0.858–0.893)	<0.0001
Alcohol related diagnoses	0.957 (0.930–0.985)	0.003	1.244 (1.205–1.283)	<0.0001
Abnormal renal function	1.167 (1.138–1.197)	<0.0001	1.021 (0.995–1.048)	0.12
Lung disease	0.920 (0.904–0.936)	<0.0001	0.979 (0.956–1.003)	0.09
Sleep apnoea syndrome	0.718 (0.697–0.740)	<0.0001	0.908 (0.879–0.936)	<0.0001
COPD	0.789 (0.771–0.808)	<0.0001	0.853 (0.826–0.882)	<0.0001
Liver disease	0.893 (0.860–0.927)	<0.0001	0.991 (0.952–1.030)	0.64
Hyperthyroidism	1.114 (1.061–1.169)	<0.0001	1.133 (1.080–1.189)	<0.0001
History of hypothyroidism	1.050 (1.026–1.074)	<0.0001	0.939 (0.918–0.962)	<0.0001
Inflammatory disease	1.070 (1.043–1.098)	<0.0001	0.979 (0.954–1.005)	0.11
Anaemia	1.146 (1.127–1.165)	<0.0001	1.078 (1.059–1.096)	<0.0001
Previous cancer	0.905 (0.888–0.922)	<0.0001	0.941 (0.923–0.958)	<0.0001

CABG: coronary artery bypass graft; CI: confidence interval; COPD: chronic obstructive pulmonary disease; HR: hazard ratio; MI: myocardial infarction; ICD: implantable cardioverter defibrillator; PCI: percutaneous coronary intervention.

**Table 3 jcm-11-01342-t003:** Yearly incidence of all-cause death and cardiovascular death in AF patients according to sex and presence or absence of concomitant hyperthyroidism.

	All Patients, Hyperthyroidism	All Patients, No Hyperthyroidism	Men, Hyperthyroidism	Men, No Hyperthyroidism	Women, History of Hyperthyroidism	Women, No History of Hyperthyroidism
All-cause death						
Whole FU	12.4 (12.2–12.7)	12.7 (12.6–12.7)	12.6 (12.1–13.0)	12.6 (12.5–12.6)	12.3 (12.0–12.7)	12.8 (12.7–12.8)
Day 1–365	21.7 (21.1–22.3)	22.8 (22.7–22.9)	22.4 (21.4–23.5)	22.6 (22.5–22.7)	21.2 (20.4–22.0)	23.1 (22.9–23.2)
After Day 365	8.3 (8.0-8.6)	8.2 (8.1-8.2)	8.3 (7.9-8.8)	8.2 (8.2-8.2)	8.2 (7.9-8.6)	8.1 (8.1-8.1)
Cardiovascular death						
Whole FU	3.9 (3.7–4.0)	3.8 (3.8–3.9)	3.5 (3.3–3.7)	3.4 (3.4–3.5)	4.2 (4.0–4.4)	4.3 (4.3–4.3)
Day 1–365	6.9 (6.5–7.3)	7.2 (7.1–7.2)	6.1 (5.6–6.7)	6.4 (6.4–6.5)	7.4 (6.9–7.9)	8.1 (8.0–8.2)
After Day 365	2.6 (2.4–2.7)	2.3 (2.3–2.4)	2.4 (2.2–2.7)	2.2 (2.1–2.2)	2.7 (2.5–2.9)	2.6 (2.6–2.6)

**Table 4 jcm-11-01342-t004:** Hazard ratios for the association between hyperthyroidism and all-cause mortality and cardiovascular mortality during (1) whole follow-up, (2) the first year of follow-up after concomitant diagnoses of AF and hyperthyroidism, and (3) beyond the first year of follow-up.

	Whole FU	Day 1–365	After Day 365
All-cause mortality			
All patients	0.973 (0.952–0.994)	0.939 (0.911–0.968)	1.016 (0.983–1.049)
Men	0.997 (0.963–1.032)	0.982 (0.937–1.029)	1.014 (0.965–1.067)
Women	0.957 (0.930–0.985)	0.910 (0.876–0.946)	1.018 (0.976–1.062)
Cardiovascular mortality			
All patients	1.006 (0.967–1.046)	0.946 (0.897–0.998)	1.088 (1.027–1.153)
Men	1.015 (0.951–1.083)	0.940 (0.859–1.028)	1.110 (1.011–1.219)
Women	0.955 (0.909–1.003)	0.904 (0.846–0.965)	1.030 (0.957–1.110)

Hazard ratios are for hyperthyroidism vs. no hyperthyroidism.

## Data Availability

This study used data from human subjects, the data and everything pertaining to the data are governed by the French Health Agencies thereby cannot be made available to other researchers.

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
