# Peer review of "Yearly Incidence of Stroke and Bleeding in Atrial Fibrillation with Concomitant Hyperthyroidism: A National Discharge Database Study"

_jcm, 2022, doi:10.3390/jcm11051342_

Round 1
Reviewer 1 Report
Authors generally answered my comments, no further comments.
Author Response
Thank you for your kind input and reassurance! We really appreciate it!
Reviewer 2 Report
Dear authors
I enjoyed reading the manuscript and I agree that this is an original one dealing with an average important theme. Nonetheless, there are significant limitations in the design of the project, as pointed out, and I would like to see in the writing of the paper a topic of future studies and a clear statement of the authors proposing a better design to approach the question of this study. Numerous phrases are short running sentences and I recommend a more detailed and complete review in order to make the text reliably and more comfortably read and understandable.
Author Response
Dear reviewer,
Thank you for your suggestions. We have amended the following content in the discussion as below:
Further large-scale prospective matched-cohort studies which investigate and characterize both the short-term and the long-term stroke events of new-onset hyperthyroidism in AF patients would be desirable. Prospective trials investigating the effectiveness and feasibility of a clinical algorithm that promotes timely and accurate stroke risk assessment, initiation of OAC, and hyperthyroidism treatment for concomitant hyperthyroidism in AF patients with varying thromboembolic risk profiles could provide further guidance to clinical practice.
Please kindly note that following your advice, we have also revised the English accordingly.
Reviewer 3 Report
The study is interesting, well conducted and well presented. Many details of the research limitations have been correctly provided.
In the discussion and in the references two recent studies about this topic should be added:
Increased risk of ischemic stroke and systemic embolism in hyperthyroidism-related atrial fibrillation: A nationwide cohort study. Am Heart J. 2021 Dec;242:123-131. Risk of Thromboembolism in Non-Valvular Atrial Fibrillation With or Without Clinical Hyperthyroidism. Glob Heart. 2021 Jun 17;16(1):45. In table 1 and in table 2 I think ischemic stroke and intracranial bleeding are previous. If it's correct, please add previous. Thank you for your research.
Author Response
Dear reviewer,
Thank you so much for your advice and recommended references.
We have added and amended the following content in the discussion section:
(page 15, paragraph 1) Similarly, a recent Korean national database-derived study also identified an increased risk of thromboembolism within the first year of hyperthyroidism-related AF diagnosis[12].
(page 15, paragraph 2) Two recent retrospective nationwide database-derived studies using propensity score-matched analyses had contradictory findings in the risk of thromboembolism associated with hyperthyroid AF[12, 21]. Cautious interpretation of these findings, while acknowledging the inherent limitations and bias of retrospective analysis of coded information is advised. Further large-scale prospective matched-cohort studies which investigate and characterize both the short-term and the long-term stroke events of new-onset hyperthyroidism in AF patients would be desirable. Prospective trials investigating the effectiveness and feasibility of a clinical algorithm that promotes timely and accurate stroke risk assessment, initiation of OAC, and hyperthyroidism treatment for concomitant hyperthyroidism in AF patients with varying thromboembolic risk profiles could provide further guidance to clinical practice.
Following your kind advice, we have also added “previous” to Table 1 and Table 2.
We really appreciate your review and input!